# REASSESSING NUMBER-DETECTOR UNITS IN CONVOLUTIONAL NEURAL NETWORKS

## ABSTRACT

Convolutional neural networks (CNNs), including CORnet, have become essential models for predicting neural activity and behavior in visual tasks. However, their ability to capture complex cognitive functions, such as numerosity discrimination, remains a topic of debate. Numerosity—the ability to perceive and estimate the number of items in a visual scene—is believed to be represented by specialized 'number-detector' units within CNNs. In this study, we utilize CORnet, a specialized type of CNN inspired by brain anatomy, which also effectively captures the variance in human behavioral data, to address the limitations of classical representational similarity analysis (RSA), which assumes equal importance for all features. We apply pruning, a feature selection technique that identifies and retains the most behaviorally relevant units. Our results demonstrate that number-detector units are not critical for population-level representations of numerosity, challenging their proposed significance in previous studies. These results can have implications for both machine learning and neuroscience.

## 1 INTRODUCTION

Early breakthroughs in the study of biological vision served as the foundation for convolutional neural networks (CNNs; Lindsay, 2021). Like the brain, these hierarchical models consist of several feedforward layers, with each layer comprising numerous artificial units that mimic neurons. Since then, CNNs have evolved into state-of-the-art models for predicting neural activity and behavior in visual tasks (Khaligh-Razavi & Kriegeskorte, 2014; Yamins et al., 2014; Yamins & DiCarlo, 2016; Cichy et al., 2016). For example, it has been demonstrated that CNNs trained on an object classification task can account for the brain responses of both humans' and monkeys' inferior temporal cortex (IT), a key region for object recognition (Khaligh-Razavi & Kriegeskorte, 2014).

But what happens when the images contain multiple objects? Perceiving and representing the number of items in a set—known as numerosity—without counting is considered a core and ancient cognitive ability shared by humans and many animal species, often referred to as 'number sense' (Dehaene, 2011). Specialized neurons, or 'number neurons', that are tuned to the number of items in a visual display have been identified in numerically naive monkeys (Viswanathan & Nieder, 2013), crows (Wagener et al., 2018), and untrained 10-day-old domestic chicks (Kobylkov et al., 2022), suggesting that numerosity is automatically represented in the brain. Brain imaging studies have also pinpointed regions in the parietal cortex that are responsible for representing numerosity in both adults (Piazza et al., 2004; Castaldi et al., 2019; Karami et al., 2023) and preverbal infants (Izard et al., 2008; Hyde et al., 2010; Edwards et al., 2015), demonstrating this ability at the population level. Additionally, fMRI decoding in the parietal regions of adults has been linked to behavioral numerosity discrimination acuity (Lasne et al., 2019). Collectively, these findings highlight the critical role of parietal brain activity in human numerosity discrimination.

Recently, it has been shown that number-detector units, analogous to number neurons recorded in the prefrontal and parietal cortices of monkeys, can emerge in the final layers of CNNs trained for visual object recognition (Nasr et al., 2019) and even in completely untrained CNNs (Kim et al., 2021). However, Karami et al. (2023), using representational similarity analysis (RSA; Kriegeskorte, 2008), demonstrated that CNNs fall short of explaining the variance in numerosity representation observed in fMRI data from the human parietal cortex. Further analysis using multidimensional scaling (MDS; Kruskal, 1964) revealed significant differences in the geometric structure of numerosity

representations between human parietal regions and CNNs (Karami et al., 2023). In the classical RSA framework used by Karami (2024), all features contribute equally to the final dissimilarity estimate. However, this 'equal weights' assumption conflicts with the notion that, when comparing representational dissimilarity matrices (RDMs), certain features may carry more informative content than others. As a result, this approach can underestimate the true correspondence between the model and a specific brain region or behavior (Kaniuth & Hebart, 2022; Tarigopula et al., 2023). Moreover, the classical RSA approach may overemphasize non-relevant units by treating them as equally important as units that carry behaviorally relevant information, such as number-detector units in our case.

To assess the relevance of number-detector units in representing numerosity at the population level within CNNs, we employed a pruning approach. Pruning is a feature selection technique used to identify and retain the most relevant parts of a model, such as specific weights or activations, that best align with the behavior data and improve predictions (Flechas Manrique et al., 2023; Bao & Hasson, 2024; Truong et al., 2024). This approach is based on the observation that pretrained models often contain redundant information (Cheng et al., 2015; Frankle & Carbin, 2018). Therefore, using the entire model may not be necessary for a specific task, such as numerosity discrimination in our case. Specifically, we pruned different CNN architectures based on the number RDM, which captures the behavioral signature of numerosity perception in humans. This matrix serves as a benchmark for comparing the alignment of CNN representations with human numerosity processing. Our results revealed that number-detector units are not critical for representing numerosity at the population level within these networks. Furthermore, we demonstrated that the semantic embeddings of written numbers extracted from GloVe (Pennington et al., 2014)—a simple distributional semantic model, which has been shown to predict human performance in numerosity comparison tasks (Rinaldi et al., 2022; Ren & Libertus, 2024)—better explain the variance in activity from pruned units compared to number-detector units. Together these findings suggest that while number-detector units may emerge in specific layers of CNNs, they do not play a significant role in capturing the broader, population-level representation of numerosity, as reflected in human behavioral data.

## 2 METHODS AND EXPERIMENTAL SETUP

### 2.1 STIMULI AND TRAINING THE CNN

To investigate number-detector units in CNNs we used CORnet-Z and CORnet-S, models with four anatomically mapped areas (V1, V2, V4, and IT) followed by a decoder layer. CORnet-Z is the simplest network in the CORnet family and a lightweight alternative to AlexNet. CORnet-S also has recurrent connectivity and is designed to maximize Brain-Score, which measures the alignment of artificial neural networks with human brain activity and behavior (Schrimpf et al., 2018). Each anatomically mapped area in the CORnet consists of a single convolution, followed by a ReLU nonlinearity, max pooling and the decoder is a 1000-way linear classifier (Kubilius et al., 2018; 2019).

We chose CORnet-Z and CORnet-S because it balanced the resemblance to the architectures used by previous studies on numerosity and because it well fit the visual system. We used three versions of the CORnet:

1. the completely untrained version with randomly initialized weights to reveal the effect of architecture alone (Cichy et al., 2016),

2. a version trained on object recognition using the ImageNet dataset (Deng et al., 2009), which contained 1.2 million images of objects over 1000 categories, as it has been used in a previous study by Nasr et al. (2019), and

3. a version of the network was specifically trained to discriminate between ten numerosity values: 6, 7, 9, 10, 12, 14, 17, 20, 24, and 29. We specifically trained the networks to discriminate between numbers because Mistry et al. (2023) demonstrated that training a CNN for numerosity discrimination significantly reorganizes the number-detector units. To avoid flawed stimulus design (Park, 2022), where low-level visual features like size or dot density correlate with the number of dots, we used the method introduced by DeWind et al. (2015) to generate the dot sets. A sample of the generated stimuli used for training the network is shown in Figure 1A. Following the approach of Mistry et al. (2023), we first initialized the network with weights pre-trained on

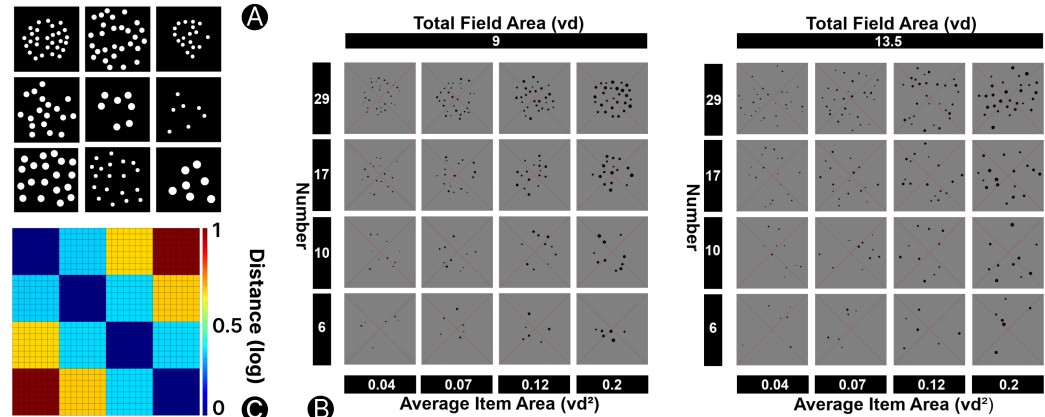

Figure 1: (A) Sample stimuli used for training the networks. (B) Sample stimuli used for testing the networks. (C) Number RDM, reflecting human behavioral data, used to prune the network layers.

ImageNet, then trained it on the numerosity task for 100 epochs using the Stochastic Gradient Descent (SGD) optimizer with default PyTorch parameters. The code used to train the networks, along with a link to the trained network weights, will be available in the GitHub repository associated with this paper after the anonymous review process.

After training, the three versions of the model were tested with visual dot sets, where both the number of dots and low-level visual features (average item area and total field area) varied across 32 different conditions: 4 numerosities (6, 10, 17, 29), 4 average item areas, and 2 total field areas (Figure 1B). Each input image was 500 × 500 pixels. We selected four layers of the network-analogous to visual brain areas (V1, V2, V4, and IT) and extracted the activations of all nodes in each layer. The results from 100 instances of each condition were averaged to produce a single activity vector for each condition from the output of each layer.

## 2.2 Identification of Number-Detector Units

Following Nasr et al. (2019) and Kim et al. (2021), we used ANOVA to find number-detector units. Specifically, a three-way analysis of variance (ANOVA) with three factors - numerosity, total field area, and average item area - was applied to select number-detector units. The goal was to identify units that had a significant change in response across different numerosities while maintaining an invariant response across variations in total field area and average item area, as well as across the three interactions among pairs of factors, and the interaction among all of the three. A unit is marked as a number-detector if it shows a significant change for numerosity ($p < 0.01$) but no significant change for the other two factors or any of the interactions. Units that did not meet these criteria were classified as non-selective. This method of selecting number-detector units is analogous to the method that has been used to detect numerosity-sensitive neurons in monkeys and humans.

## 2.3 Representational Similarity Analysis between RDMs

To create the CNNs' RDMs, we used the 32 activity vectors obtained by averaging the 100 instances per condition. We chose 100 instances to address concerns about the limited number of sample images used in previous studies, such as Nasr et al. (2019), which were criticized for this limitation (Zhang & Wu, 2020). The CNNs' RDMs were constructed using 1 − Pearson correlation between the activations of each layer for each pair of conditions. The number RDM (Figure 1C) was based on the logarithmic distance between the pairs of conditions in terms of numerosity. RDMs from GloVe embeddings were created using 1 − Pearson correlation of 300-dimensional embeddings between the written forms of four numbers: '6', '10', '17', and '29'. To achieve the final RSA scores, we extracted the lower triangular portion of each RDM, excluding the diagonal, and vectorized them. These vectors, referred to as representational dissimilarity vectors (RDVs), were then used to compute the Pearson correlation between each CNN's RDV and the number RDV.

### 2.3.1 PRUNING THE LAYERS OF MODELS

The pruning algorithm, which is adapted from Tarigopula et al. (2023), involves three steps. First, the importance of each unit is assessed by removing it from the full set of units. Each time a unit is removed, a new RDM is computed, and its score is compared to the number RDM. A significant drop in the score compared to the full set RDM indicates that the unit is important for matching the number RDM, while a smaller drop or an increase in score suggests the unit is unimportant or possibly encoding noise. Second, all units are ranked based on their importance scores, from highest to lowest. Third, starting with an empty activation vector, units are sequentially added back in the order of their ranking. After each addition, the fit between the RDM derived from the new embedding and the number RDM is re-evaluated. We truncate and select the set of neurons when the highest RSA score is achieved, and refer these units as the 'retained units' after pruning.

It is important to note that the pruning procedure was guided by the number RDM (figure 1C), not the GloVe RDM, as our goal when introducing GloVe RDM is to assess how well pruning generalizes from the vision domain to the language domain. The GloVe RDM serves as an evaluation rather than a supervised signal for selecting units.

## 3 RESULTS

### 3.1 RETAINED UNITS AFTER PRUNING AND NUMBER-DETECTOR UNITS OFTEN DO NOT OVERLAP

Table 1: Number of retained units after pruning, and of number-detector units identified by ANOVA. The full set of units in V1, V2, V4, and IT layer in both models are 262144, 131072, 65536, 32768 respectively. The numbers in parentheses denote the percentage of units compared to the full set.

| CORnet | Layer | After Pruning | | | ANOVA | | |
|---|---|---|---|---|---|---|---|
| | | ImageNet | DeWind | Untrained | ImageNet | DeWind | Untrained |
| Z | V1 | 101511 (39) | 100081 (38) | 72429 (28) | 68 (0.03) | 83 (0.03) | 77 (0.03) |
| | V2 | 45421 (35) | 45584 (35) | 40176 (31) | 44 (0.03) | 24 (0.02) | 1 (0.00) |
| | V4 | 32264 (49) | 31979 (49) | 20358 (31) | 13 (0.02) | 27 (0.04) | 13 (0.02) |
| | IT | 6074 (19) | 1971 (6) | 4139 (13) | 32 (0.1) | 10 (0.03) | 3 (0.01) |
| S | V1 | 58310 (22) | 117169 (45) | 100637 (38) | 803 (0.31) | 939 (0.36) | 657 (0.25) |
| | V2 | 6886 (5) | 39881 (30) | 12344 (9) | 414 (0.32) | 121 (0.09) | 454 (0.35) |
| | V4 | 423 (1) | 16444 (25) | 2595 (4) | 101 (0.15) | 62 (0.09) | 257 (0.39) |
| | IT | 66 (0.2) | 37 (0.1) | 424 (1.3) | 126 (0.38) | 0 (0) | 125 (0.38) |

Table 2: The overlap scores between the units selected by the two methods. The scores are defined as the number of units presented in both two sets divides by the number of units in the smaller set, ranging from 0 (in case of no overlap) to 1 (in case of the bigger set contains the entire smaller set). The value for CORnet-S trained on DeWind was undefined as there was no unit detected via ANOVA.

| CORnet | Layer | ImageNet | DeWind | Untrained |
|---|---|---|---|---|
| Z | V1 | 0.40 | 0.57 | 0 |
| | V2 | 0.59 | 0.71 | 1 |
| | V4 | 1 | 0.85 | 1 |
| | IT | 0.09 | 0 | 0 |
| S | V1 | 0.31 | 0.27 | 0.65 |
| | V2 | 0.01 | 0.21 | 0.01 |
| | V4 | 0 | 0 | 0 |
| | IT | 0 | - | 0 |

Table 1 presents the number of units selected by two methods: pruning and ANOVA. In both models, the number of retained units after pruning is significantly higher than the number of number-detector units identified by ANOVA. ANOVA results in significant more units in CORnet-S compared to Z. Moreover, pruning removed the largest proportion of units in the IT layer, while the ANOVA method showed no significant differences across layers. There is no noticeable difference between

the ImageNet-trained and DeWind-trained models, suggesting that there are no advantages on reducing the dimensions when training the models on numerosity discrimination tasks beforehand. Interestingly, both retained units and number-detector units are found in untrained networks, consistent with the findings of Kim et al. (2021). However, in the untrained models, pruning often retained more units in CORnet-Z compared to S, while ANOVA resulted in the opposite pattern.

The overlap between the units selected by the two methods is defined as the number of units shared by both sets, divided by the number of units in the smaller set: |Pruning ∩ ANOVA| / min(|Pruning|, |ANOVA|). The overlap scores, as shown in table 2, vary considerably across layers and models. Little to no overlap was observed in the IT layer of both models and in the V4 layer of CORnet-S, while significant overlap was found in the V2 and V4 layers of CORnet-Z, and in the V1 layer of CORnet-S. Only three cases showed a perfect overlap score of 1, while seven cases had a score of 0. Overall, the results suggest that the two methods generally select non-overlapping sets of units.

### 3.2 Retained Units after Pruning Fit the Behavior Data Better than Number-Detector Units

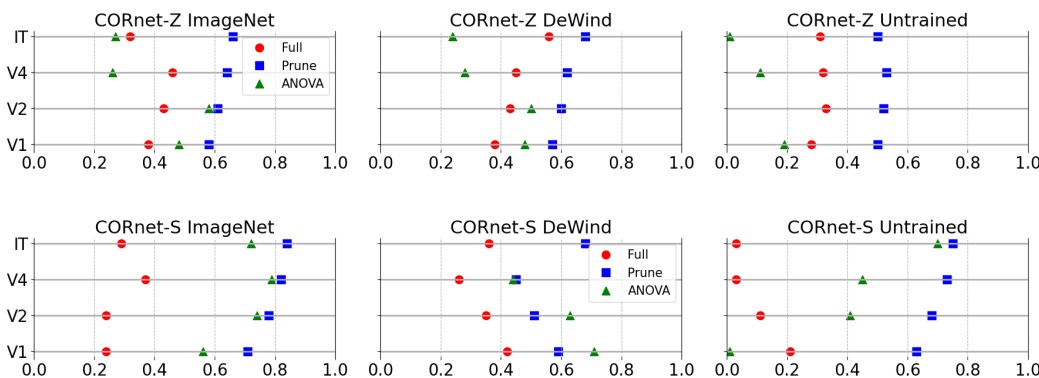

Figure 2: The Pearson correlations, quantified using RSA, between the RDMs derived from the full set of units, the retained units after pruning, and the number-detector units identified by ANOVA, vs. the number RDM. The missing data points are due to an insufficient number of minimum units (2) required to compute the correlations.

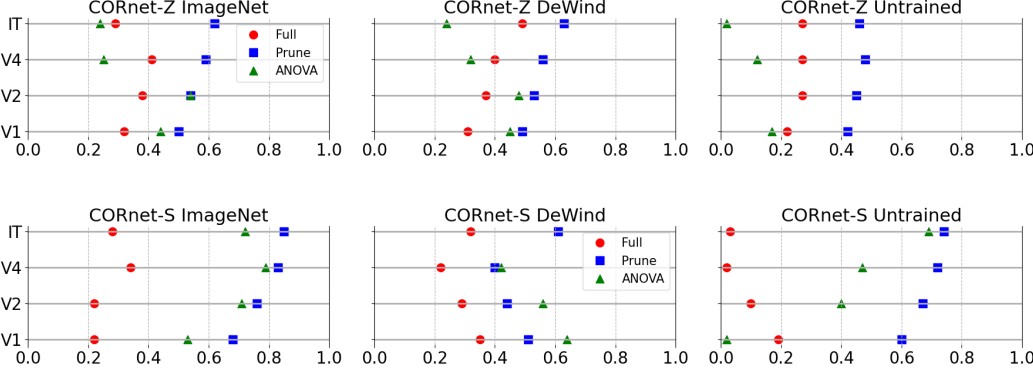

Figure 3: The Pearson correlations, quantified using RSA, between the RDMs derived from full and selected units, vs. the GloVe RDM.

Figure 2 shows that the retained units after pruning provide a better fit for modeling the number RDM compared to the full set of units, highlighting the limitations of using the full set in classical RSA. Additionally, in most cases—except for V2 and V1 in CORnet-S DeWind, which is specifically

trained for number discrimination—the number-detector units selected by ANOVA, as commonly studied in traditional literature, perform worse than the retained units after pruning. In 8/22 cases, they are even worse than the full set of units.

Regarding the fit between selected units and language representations (figure 3), the ranking of methods remains mostly consistent with the results of number RDM, while there is a slight decrease in score magnitude. This is expected, given the high correlation between the number RDM and GloVe RDM (Pearson correlation 0.88). In summary, this suggests that number-detector units are not important for capturing the population-level representation of numerosity in human behavioral data and do not provide a strong baseline for alignment with the language domain.

## 4 DISCUSSION

It has been emphasized that cognitive and behavioral functions emerge from the collective dynamics of neural populations, rather than isolated neuronal activity (Yuste, 2015), underscoring the importance of population-level analysis for understanding complex behaviors like numerosity discrimination. In this context, RSA has become a widely used method for comparing representational spaces from human brain activity, behavioral data, and computational models. However, classical RSA assumes that each feature is equally important, making it difficult to interpret the contribution of individual features. In our case, classical RSA does not provide insights into the importance of number-detector units identified in CNNs. To address this limitation, we applied pruning, a feature selection technique used to identify and retain the most relevant components of a model—such as specific weights or activations—that best align with behavioral data.

Pruning revealed that almost none of the number-detector units contribute to the representation of numerosity at the population level, casting doubt on their significance for numerosity discrimination. These findings are also consistent with previous work by Mistry et al. (2023), which demonstrated that training a CNN for numerosity discrimination significantly reorganizes the number-detector units, and by Chapalain et al. (2024), which showed that dot-pattern-tuned units do not generalize to object-number information in photorealistic stimuli. Furthermore, we demonstrated that the pruned units more effectively explain the variance in the GloVe embeddings of the written form of numbers. This result revealed that pruning can better align the visual and written forms of number representation.

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
