# OpenReview forum: "Reassessing Number-Detector Units in Convolutional Neural Networks"
_ICLR.cc/2025/Conference — ICLR 2025 Conference Withdrawn Submission_

### Official Review · Reviewer_A5bV · 2024-10-26

**Soundness:** 2
**Presentation:** 2
**Contribution:** 2
**Rating:** 5
**Confidence:** 3

**Summary:**

This work uses network pruning over classical RSA to identify and retain the most relevant units of a network for the task of numerosity. Pruning is based on behavioral signatures of numerosity perception in humans. The authors find that almost no number-detector units contribute to population-level representations of numerosity, when evaluated using two models—CORnet-Z and CORnet-S. Additionally, semantic embeddings of written numbers extracted from GloVe explains variance in activity from pruned network units better than number-detector units.

**Strengths:**

- The manuscript is well written, and generally easy to follow.
- The authors try to resolve the issue with using RSA for understanding the contribution of number-detector units in CNNs through a pruning-based scheme.
- The authors identify that number-detector units do not seem to contribute to the representation of numerosity at the population level, in contrast to what some of the previous works have established.
- Pruning is presented as a way to better align visual and written forms of number representations.

**Weaknesses:**

I would appreciate if the authors could elaborate on the scope of their contributions. Currently, as it seems to me, a pruning-based method is used to identify “neurons” that align best with behavioral data. As the authors cite, there is rich literature on the topic already. The novelty, it seems, is in the application of this scheme to the singular task of numerosity perception and the insights that can be gathered from doing so. However, insights come from only two deep artificial neural networks, namely CORnet-Z and CORnet-S (i.e., a single family of model classes). While CORnet-S is a decent predictor of neural activity and image-by-image human behavior, I would be more convinced by the results when a breadth of model classes such as AlexNet, ViTs, ResNets, etc. are used to support the claims made. This becomes particularly important since none of the models truly replicate the ventral visual cortex, but are mere models of it to some degree.

Additionally, CNNs, as was quoted in the manuscript, contain redundant information encoded in its units. Then, how do we synthesize this observation with the authors having used pruning to identify units that increase correlation with the number RDM, wherein dropping units that are not necessarily numerosity-coding still result in a higher score (i.e., ability to predict the number RDM) upon being pruned (what we could call artifacts)? For example, it might be possible that removing units that encode noise result in a significantly higher score (akin to network regularization).


**Things (manuscript writing) that can be improved but did not impact my score:**
Line 214 - “ANOVA results in significantly more units in CORnet-S …” (typo).

**Questions:**

Lines 165-6: “Each time a unit is removed, a new RDM is computed, and its score is compared to the number RDM”. Are significance scores corrected for multiple comparisons here?

---

### Official Review · Reviewer_Cz2b · 2024-10-29

**Soundness:** 2
**Presentation:** 1
**Contribution:** 2
**Rating:** 3
**Confidence:** 4

**Summary:**

This paper reexamines the role of "number-detector" units in convolutional neural networks (CNNs) with respect to numerosity discrimination tasks. Using CORnet models (CORnet-Z and CORnet-S), which are CNNs inspired by brain anatomy, the authors challenge previous assumptions about the importance of specialized number-detector units.

The key contributions are:

* Adapts a pruning-based approach (Tarigopula et al 2023) to identify behaviorally relevant units in CNNs for numerosity tasks, as an alternative to traditional ANOVA-based identification of number-detector units.

* Demonstrates that pruned units better capture population-level representations of numerosity compared to previously identified number-detector units, as measured by alignment with human behavioral data.

* Shows that the units retained after pruning have minimal overlap with traditional number-detector units (ANOVA), suggesting that number-detector units may not be critical for numerosity discrimination at the population level.

* Validates their findings across different training conditions (untrained, ImageNet-trained, and numerosity-trained networks) and demonstrates that the pruned representations also generalize well to semantic number representations from language models (GloVe).

This work provides new insights into how CNNs process numerosity information and challenges existing views about the role of specialized number-detector units in artificial neural networks, with potential implications for both machine learning and neuroscience (behavior / psychophysics).

**Strengths:**

**Originality**

* Adapting Tarigopula et al (2023) pruning strategy to retain units that best align with human behavioral data
* Demonstrating that retained units after pruning are fundamentally different from those identified by traditional ANOVA-based methods
* Establishing that pruning-retained units show stronger alignment with both human behavioral data and GloVe embeddings compared to ANOVA-identified number-detector units

**Significance**

Numerosity is an abstract concept that humans and animals master (this is well documented in the introduction) and that previous research in CNNs interpretability has attempted to explain. While previous approaches have focused on unit-level analysis, this work provides an alternative perspective by examining population-level properties of numerosity in deep networks. A key contribution is the novel use of human behavioral data ("the number RDM") to guide the pruning strategy, thereby identifying units that best align with human representations.

The use of the number RDM makes this work relevant for ICLR in the domains of representation learning and human-aligned AI systems.

**Weaknesses:**

While the introduction is quite extensive and well-written, the paper appears to have been written in haste and lacks proper organization (only 6 pages out of 9/10 have been used; figures could have been larger and more detailed). Here's a list of possible improvements and weaknesses.

**Major Weaknesses**

* The number RDM plays a central role in your paper, but its origin is unclear. There are no references supporting it in the text or in Figure 1. Adding labels to the number RDM and including a reference in the figure caption would make it clearer and easier to understand.

* Although the comparison with GloVe aims to strengthen the claim of population-level representations for numerosity, it represents too much of a paradigm shift. GloVe inherently learns context through word-word co-occurrence training. Furthermore, word representation in a language model differs significantly from counting in visual representations (see K. Erk, "What do you know about an alligator when you know the company it keeps?", Semantics & Pragmatics Vol 9, 2016). One potential improvement would be to run a control experiment randomizing the written forms of numbers, as specific number representations might be naturally close to each other. Comparing textual and visual representations is risky, especially when inductive biases differ and representations are fundamentally different.

* Regarding the GloVe embeddings, seeing the GloVe RDM would be helpful. The comparison method with CORNet's RDMs is unclear: GloVe has 4 conditions (presumably the written forms of numbers 6, 10, 17, and 29), while CORNet has 32 conditions (4 numerosities, 4 average item areas, and 2 total field areas). Is the comparison limited to one set of 4 numerosities (with fixed average item areas and total field areas)? If so, which set was used, and what happens when varying the other two conditions? The text doesn't adequately address these questions.

**Minor Weaknesses**

* Table 2 presents overlap scores between units selected by two methods. For ANOVA, the number of selected units is very small (sometimes just 1 unit), making it difficult to interpret overlap scores without a null model. What would be the overlap score of a random model - for instance, the largest set selected randomly among the network's neurons per layer (using the same percentage as post-pruning)? Since the post-pruning percentages significantly exceed the ANOVA ones, the random overlap would likely differ from zero.

* In subsection 2.1, you state that each image was 500 x 500 pixels. Since CORNet typically uses ImageNet-shaped inputs (224x224 RGB), did you perform any resizing, cropping, or other operations on the original 500x500 pixel images? If so, which operations were used? How might this affect the average item areas and total field areas conditions?

* The introduction lacks historical or established review references to pruning, such as:

     **Historical**
     - Janowsky, "Pruning versus clipping in neural networks", Physical Review A, 1989
     - Mozer and Smolensky,  *Using Relevance toReduce Network Size Automatically*. Connection Science 1989
     - Karnin, *A simple procedure for pruning backpropagation trained neural networks.*  IEEE transactions on neural networks, 1990

    **Review**
     - Blalock et al., "What is the State of Neural Network Pruning?", Proceedings of Machine Learning and Systems, 2020

**Questions:**

Some questions have already been addressed in the **Weaknesses** section, but I have a few more:

* Is classification performance maintained after pruning? Is there a significant reduction in performance? Typically, pruning is used to remove redundant information while maintaining (and sometimes even improving) classification accuracy.

* Since it's not possible to understand what the number RDM represents (no reference provided, see weaknesses), I assume it's behavioral data from a psychophysics experiment, as suggested in the text. If this is the case, what is the rationale for using an RDM derived from psychophysics experiments to prune layers that supposedly correspond to V1, V2, V4, and IT (CORNet)? For example, I would expect it to be relevant for the readout layer, which is typically matched to behavioral benchmarks in Brainscore.

---

### Official Review · Reviewer_TZr9 · 2024-11-04

**Soundness:** 2
**Presentation:** 2
**Contribution:** 1
**Rating:** 3
**Confidence:** 4

**Summary:**

This paper trains neural network on number detection and shows how RSA-based pruning identifies different sets of artificial neurons than the "number detectors" found in the network.

**Strengths:**

The paper compares in detail of the numerosity detector methods and the RDM analysis in identifying number representations in networks. The results could be helpful in merging conflicting evidences people found with these methods. The writing is relatively clear.

**Weaknesses:**

The findings are insufficient in this paper:
- The results shown in Figure 2 seem circular to me. If the pruning is guided by RSA scores with the number RDM then it is very much expected that pruned network will get the best correlation scores.
- For Table 1 and 2, it is unclear how robust these numbers or patterns are without different network initialization/training to show confidence interval on these numbers
- It very well could be the case that the network develops both population coding and detector units to represent numbers. I don't think the findings here are surprising to me. Perhaps a better way to asses what kind of representation are the ones we want to study, is to ablate the units identified from each method and see how they impair network performance in detecting number.
- The results presented here lacks impact as well. Aside from above results, the paper lacks a main finding that speaks to a larger audience in this conference. The results reported here seem to be only possibly useful to those who study number representation with neural networks. If the main thesis is how RSA or pruned RSA units does not align with detector neurons, perhaps a wider range of modality and task could be used to further prove this point for object or other visual representations and this result will be of interest of broader computational neuroscience community.

**Questions:**

- Is the identification of number detector neurons picking out neurons that are sensitive to a specific number or does it also pick out neurons that are selective for a few numbers (e.g. 6 & 10 but not 17 &19). Or in another words, are you considering a graded representation of numerosity? If it is the former case, would relaxing the criteria to non-specific number detector lead to different results?
- What is considered significant RDM drop (line 167)?
- Are the large number in untrained network left after pruning concerning?
- Maybe I am missing the citation for this, but where is human behavioral data from? How do we know that human are invariant to total field and average item area?

---

### Note · Authors · 2024-11-17

**Comment:**

After consideration of all the reviewers' feedback, we decided to withdraw our work to have more time for improvement. We thank the reviewers for their time and effort in evaluating our submission.

**Withdrawal Confirmation:**

I have read and agree with the venue's withdrawal policy on behalf of myself and my co-authors.